# The Study of Health-Related Fitness Normative Scores for Nepalese Older Adults

**DOI:** 10.3390/ijerph17082723

**Published:** 2020-04-15

**Authors:** Jung Kyu Kim, Won Il Son, Ye Jung Sim, Ju Sung Lee, Kamala Oli Saud

**Affiliations:** 1Exercise Physiology Lab, Department of Leisure Sports, College of Humanities, Social Science and Design, Sports, Kangwon National University, 346 Jungang-ro, Samcheok-si, Gangwon-do 25913, Korea; jkkim67@kangwon.ac.kr (J.K.K.); syjim22@kangwon.ac.kr (Y.J.S.); 2Physical Education Measurement and Evaluation Lab, Department of Leisure Sports, College of Humanities, Social Science and Design, Sports, Kangwon National University, 346 Jungang-ro, Samcheok-si, Gangwon-do 25913, Korea; son91@kangwon.ac.kr; 3Department of Sports Sciece, Kangwon National University, 1 Kangwon Daehaggil, Chuncheoun-si, Gangwon-do 24341, Korea; jsleetime@kangwon.ac.kr

**Keywords:** older adults, health-related fitness test, evaluation norms, awards norms, Nepal

## Abstract

Physical fitness tests are important to maintain and promote the health status of people. The purpose of this study was to develop health-related fitness evaluation norms according to the age and gender of Nepalese older adults. One thousand nine subjects (449 males, 560 females) above 60 years, residing in 19 wards (rural and urban) of Dhangadhi Sub-Metropolitan City participated in this study. The test included the PAR-Q (Physical Activity Readiness Questionnaire), social aspects questionnaire, blood pressure test, height, weight, BMI (body mass index), percent body fat, and four physical fitness components (grip strength, 1-minute sit-to-stand, sit and reach, and 2-minute step tests). Mean, SD, and fitness evaluation norms for each component were obtained after the main test and statistical analyses. This study showed higher BMI and percent body fat in female age groups than in male age groups. Grip strength, relative grip strength, sit-to-stand, and 2-minute steps scores were better in male age groups than in female age groups, but in contrast, flexibility was better in female age groups. This study may help the related sectors to assess physical fitness, identify fitness levels, and develop appropriate physical activities or exercise programs for older adults based on age.

## 1. Introduction

Aging is a natural phenomenon and an inevitable process. Population ageing has been considered as one of the important demographic phenomena that impact directly or indirectly on health-care expenditure and the kinds of formal and informal care services. Developing nations like Nepal have not yet succeeded in managing appropriate health-care systems and establishing physical activity to ensure health and a high quality of life for older adults.

Nepalese society is in a phase of modernization that is changing the age structure of the country with a shift towards older ages through declining fertility, relatively controlled adult mortality [1], and improved health interventions [2]. According to the national census, there were 1.5 million people in Nepal over 60 years of age in 2001 and 2.1 million in 2011, which constitutes 6.5% and 8.1% of the total population in the country, respectively [3,4]. During the years 1991–2001, the annual elderly population growth rate was 3.39%, higher than the annual population growth rate of 2.3%, which indicates the start of aging dynamics in Nepal [1,5]. Because of this, declining health is an issue. It is imperative to examine quantifiable outcomes of habitual physical activity if the contribution of physical activity to the health of the aging population is to be better understood.

Physical fitness is the ability of body systems that function efficiently to perform activities of daily living, respond in emergency situations, and enjoy sports and leisure activities [6]. It is one of the important indices of health that helps people to carry out daily activities safely and independently without undue fatigue [7]. A new fitness concept, health-related fitness, has been introduced on the basis of relationships between physical activities, fitness, and health [8]. Health-related fitness refers to the components of fitness (cardiorespiratory endurance, muscular strength, muscular endurance, and flexibility) [7] that are affected by habitual physical activity and are related to various health outcomes. Lower levels of fitness are associated with all-cause mortality, a high risk of cardiovascular disease, cancer, functional disability, and arterial stiffness [9,10,11,12,13,14,15,16]. Individuals having an appropriate level of health-related fitness may work efficiently, reduce the risk of disease and injury, look their physical best, and participate and enjoy physical activity [17]. Fitness, in various forms, increases performance and improves safety [18] and leads to greater reserve capacity to resist physical stress [19]. Improvements in physical fitness through appropriate physical activities in an older adult should enable them to maintain and promote health and well-being [20,21] into later years of life.

Physical fitness testing is a highly visible and important part of physical fitness programs. It is important to monitor fitness to evaluate risk factors for health hazards in older adults and to ensure the safety of starting new activities. In Nepal, information on the health condition of older adults is still insufficient and no study on health-related fitness tests of older adults was found in the literature. The purpose of this study was to develop health-related fitness norms according to age and gender of older adults, with the expectation that our research might provide useful information for targeting effective physical activity interventions towards people at risk for declining health and to improve the health status of the general population.

## 2. Materials and Methods 

Prior to the study commencing, approval for the research with an ethical code 2076/077 was obtained by the Dhangadhi Sub-Metropolitan Urban Health Centre and Dhangadhi Sub-Metropolitan City, Nepal. Thirty volunteers were recruited in the combined efforts of the Sub-Metropolitan City and Sudur Paschimanchal Campus. This study was conducted from the last week of July to the third week of August in 2019. Four days of training were provided to the volunteers. A permanent health assistant for the study period was provided by the Urban Health Centre, Dhangadhi Sub-Metropolitan City. A local health assistant and volunteers were provided by the health post of each ward of the Sub-Metropolitan City. Advertising was run through local FM radio and TV spots in the representative sectors sharing information about the test, dates, places, and times starting a week before the tests commenced. On test day, after fully briefing participants about the nature and purpose of the study, a completed Physical Activity Readiness Questionnaire and written informed consent were received from each participant. Completion of the social aspects questionnaires and overseeing of the tests were performed by trained volunteers. Before each test, a trained volunteer explained and demonstrated the task and made sure that the task could be completed without any physical risk to the participant. Volunteers helped the participants in doing a warm-up consisting of head turns and half circles, single arm crossovers and chest stretches, and calf and hamstring stretches after the body composition tests and before the physical fitness tests [22]. The test unit was always prepared for immediate first aid for injuries and sudden illness during the tests. 

### 2.1. Characteristics of the Fitness Test

Physical factors necessary to carry out the activities of daily living were included. The test included mainly health-related fitness factors (body composition, muscle strength and endurance, flexibility and cardiorespiratory endurance). It was intended to be field-based and low cost. Fitness test components were selected for having a low risk of injury during testing.

### 2.2. Participants

A field-based cross-sectional study including 1009 subjects, (449 males, 560 females) above 60 years, residing in 19 wards (rural and urban) of Dhangadhi Sub-Metropolitan City participated in the health-related physical fitness test. Participants completed the blood pressure test after resting 5–10 min. Participants with any problems identified in the Physical Activity Readiness Questionnaire (PAR-Q); having systolic pressure greater than 160 mmHg and diastolic pressure greater than 100 mmHg; or having an implantable electrical device such as a pacemaker, defibrillator, or nerve stimulator were excluded from the other tests. 

After completing the blood pressure test, the five remaining tests were completed in the following order to minimize fatigue: body composition analyses, 1-min sit-to-stand, grip strength, and flexibility [22].

### 2.3. Body Composition Tests

The Body Composition Analyzer (Inbody 470, Korea) was used to assess weight, body mass index (BMI), and percent body fat. BMI is body weight in kilograms (kg) divided by height in meters squared (m^2^). Percent body fat is the total mass of fat divided by total body mass and multiplied by 100, which includes essential body fat and storage body fat. A stadiometer was used to measure height.

### 2.4. Hand Grip Strength Test

A digital grip strength dynamometer (TKK 5401; Takei, Tokyo, Japan) was used to assess upper body strength. Participants were asked to stand and hold the dynamometer by their testing hand, which was held slightly away from the other parts of the body, and with the grip meter indicator facing outward. The measurement was recorded after squeezing the handle of the dynamometer as hard as they could without swinging the dynamometer [23].

### 2.5. Sit and Reach Test

A trunk flexibility box was used to assess body flexibility. Participants were asked to sit with bare feet, legs extended, toes pointing up, and feet approximately hip-wide, and with the soles of the feet against the base of the measuring device. Then, they were asked them to push the slide slowly forward, as far as they could, by placing one hand on top of the other and without lifting their knee from the ground. The measurement was recorded while in a static position for a couple of seconds at the point of greatest reach [24].

### 2.6. Sit-to-Stand Test

A standard chair without arm supports was used to assess lower body strength and muscle endurance. Participants were asked to stand up from and sit down on the chair with their feet resting on the floor and their hands crossed at the wrists and held stationary on their shoulders; they were asked to repeat this procedure as many times as possible within one minute at a participant-defined pace. The participant was informed when 30 and 15 s were left in the test. The number of completed repetitions was counted considering standing as being in full extension (knee and hip extension) and sitting as being in a position with their knees at 90° flexion [25].

### 2.7. Two-Minute Step Test

A two-minute step test was preformed to assess cardiorespiratory endurance. Participants were asked to stand up straight behind poles with rubber bands. A rubber band was adjusted at the level corresponding to midway between the patella and iliac crest. The subject was then asked to march in place for two minutes, lifting their knees to the height of the rubber band. Resting or sitting on a chair placed behind the participant was allowed if needed. The participant was notified when 1 min was completed and when 30 seconds were remaining, and participants were given verbal motivation. The test was stopped after two minutes of stepping, and the total number of repetitions of the right knee reaching the rubber band level in two minutes was recorded [26].

### 2.8. Statistical Procedure

Descriptive statistical analysis was used to analyze average and standard deviation by age group and gender of older adults. The Shapiro–Wilk test was completed for the normality test. Cajori’s 5-grade evaluation norms [27] were used to develop health-related physical fitness norms for the tested components on the basis of age group and gender. Percentile was used to develop fitness awards and certificates of recognition. 

## 3. Results

The general characteristics of the participants are presented in Table 1. Mean and standard deviations for anthropometric measurements height, weight, body mass index (BMI), and percent body fat are presented in Table 2, Table 3, Table 4, Table 5 and Table 6 and the health-related physical fitness components grip strength, relative grip strength (grip strength calculation considering weight), 1-min sit-to-stand, sit and reach, and 2-minute steps are presented in Table 7, Table 8, Table 9, Table 10, Table 11 and Table 12.

Table 1 shows 55.50% of the study population were female and 45.29% were janjati. Nearly 73% of the participants were illiterate and only 0.69% were graduates. Table 1 also shows that 23.29% of the participants were widowed and 85.13% were living in joint families. Nearly 95% of study group were non-smokers and nearly 86% were non-drinkers. Nearly 70% of the participants were from rural areas of Dhangadhi Sub-Metropolitan City. Table 1 also shows that nearly 78.79% of the study population never had a health check-up and only 8.62% had a health check-up within the last six months.

As indicated in Table 2, Table 3, Table 4, Table 5, Table 6, Table 7, Table 8, Table 9, Table 10, Table 11 and Table 12, gradual declines in scores were observed over the 5-year age spans for both males and females in all the tests except height and weight in both males and females and percent body fat in males. Height was similar in both male and female age groups. Weight was similar in both males and females from the 60–64 and 70–74 year age groups but females’ average weight was higher in the 75–79 and 80+ age groups than the that of the same male age groups. BMI and percent body fat were higher in the female age groups than those in the male age groups and showed gradual declines in the female age groups as age increased. The percent body fat remained almost constant for all age groups of males. Grip strength and relative grip strength scores showed a gradual decline in the 5-year age spans and were higher in all male age groups than those of the female age groups. The result showed a gradual decline in sit-to-stand scores in both male and female age groups as age increased. Male showed better sit-to-stand scores in all age groups than those of female age groups. We also saw a gradual decline in 2-minute step scores in both male and female age groups as age increased. Male scores were better in all age groups than those of female age groups. Flexibility also declined gradually in all age groups as the age increased. In contrast with other test results, flexibility was better in all female age groups than that of male age groups. Reported diagnoses were mainly high blood pressure, diabetes, and arthritis, but some participants had heart disease.

### 3.1. Fitness Evaluation Norms

As mentioned above, Cajori’s 5-grade evaluation norms were used to develop older adults’ health-related fitness norms for the tested components on the basis of age group and gender. For the anthropometric measurements, such as BMI and percent body fat, the 5-grade relative evaluation was obesity (7%) if there was mean + 1.5σ or above and over weight (24%) if there was mean + (0.5σ~1.5σ). It was normal (38%) if there was mean ± 0.5σ. It was evaluated as thin (24%) when included in the range of mean + (−0.5σ to −1.5σ) and very thin (7%) when included below the range of mean −1.5σ. In contrast, for health-related fitness components such as grip strength, sit and reach, sit-to-stand, and 2-minute steps, the 5-grade relative evaluation was excellent (7%) if there was mean + 1.5σ or above and very good (24%) if there was mean + (0.5σ~1.5σ). It was normal (38%) if there was mean ± 0.5σ. It was poor (24%) when included in the range of mean + (−0.5σ to −1.5σ) and very poor (7%) when included below the range of mean −1.5σ (Figure 1). The evaluation norms calculated by applying Cajori’s 5-grade evaluation norms are presented in Table 13, Table 14, Table 15, Table 16, Table 17, Table 18 and Table 19.

### 3.2. Stages of Fitness Awards or Certificates for Older Adult

Recognition for achievement is considered as a motivation for those who want improvement in physical fitness. One of the main reasons for the development of the fitness awards was to provide motivation for those using the test. It is a means of self-assessment that may assure a greater degree of personal safety and the proper determination of exercise intensity. It can motivate individuals to take the test, use the test for the development of a personal strategy to improve physical fitness, and commit themselves to continue the physical activities to maintain their well-being.

Health-related physical fitness awards for Nepalese older adults was developed on the basis of the percentiles (70th, 50th, and 30th) of each test item by gender and age as shown in Table 20 [28]. Participants will get a medal award or certificate of recognition only after meeting the fitness award criteria.

Body composition scores (BMI and percent body fat) were excluded in the gold award. For the gold award, all the health-related fitness test components should be at least in the 70th percentile. Except body composition, all health-related fitness test components must be equal or above the 50th percentile and below the 70th percentile for the silver award. The bronze medal is certified if the body composition is in the recommended range. BMI must be 18 kg/m^2^ to 25 kg/m^2^ in both males and females, and percent body fat should be 7%– 35% for males and 16%– 32% for females. All four health-related fitness tests must be above the 30th percentile and below the 50th percentile.

## 4. Discussion

Maintaining independence in older adults becomes important due to the challenges of the demographic shift [29]. A fitness test is a combination of a sequence of exercises that assesses overall health and physical status of an individual. It is known to be the initial phase in the design of a suitable exercise program based on a baseline score for general health and fitness purposes [30].

In this study, we determined norms of health-related physical fitness components based on gender and age groups for older adults of Nepal. Selected test components are easy and safe to assess for inactive and very active older adults. These test components are the best to assess health-related physical fitness in a country like Nepal because they are inexpensive, space efficient, and transportable as they do not require advanced devices or separate room. 

This is the first study in the physical fitness field in Nepal. It is centered only in the single Sub-Metropolitan City of the far western development region of Nepal. The analysis shows an interesting factor that BMI data in specific age groups (60–64, 65–69, 70–74), and sit and reach data in all age groups were not in normal distribution in both genders. Further studies in every age group, children, adolescents, adults, and even in older adults, are needed. In addition, studies covering all altitudes and areas of Nepal should be completed in the future.

The basic body functions, such as strength, endurance, flexibility, and balance on extremities, are all important to sustain physical independence at older ages [31]. A reduction in lean body mass and an increase in fat mass may increase mortality risk in older adults [32]. Thus, maintaining body a mass index (BMI) at a healthy level with more muscular mass than fat mass is essential. 

Generally, fat and lean soft tissues change gradually with age and are believed to be associated primarily with imbalances between energy intake and expenditure due to an increasingly sedentary lifestyle [33]. Our study shows relative stability of percent body fat in older adult males but a decrease with age in female older adults, similar to Baumgartner et al. [33]. The World Health Organization (WHO) had recommended body mass index (BMI) greater than 23 kg/m^2^ and less than 28 kg/m^2^ as normal BMI for older adults [34]; which is higher than most BMI values found in this study. Overweight is not associated with an increased risk of mortality. In contrast, risk of mortality increases in older adults with BMI values less than 23 kg/m^2^ [32], which shows a poor health status in the older adults of Nepal.

In this study, analysis of grip strength, sit-to-stand (STS), and 2-minute steps by gender shows higher grip strength in males at all ages, and analysis by age group shows a gradual decline in grip strength in both genders, similar to other studies [25,31,35,36,37,38,39,40,41,42,43,44,45,46]. Our results show lower grip strength in Nepalese older adults than in older adults in other countries [37,38,40,41,42] and show higher relative grip strength in older females in Nepal than in Korean female older adults [42]. Relative grip strength is calculated considering weight [47], that is the reason for a higher relative grip strength in Nepalese females. The sit-to-stand test was done for 1 min similar to [25,43,44]. Nepal’s life expectancy is 70.2 years (68.8 years and 71.6 years for males and female, respectively) [48] and it is hard to find healthy 80+ older adults [49]. The 1-minute STS test is better to determine lower body muscular strength and endurance in an older adult below the age of 80 years [25]. In addition, STS performance was significantly associated with standing, good balance, coordination, and aerobic capacity that cover several components of physical fitness [43,50,51,52]. This study does not asses the balance test separately. Thus, the 1-minute sit-to-stand test was selected for lower body strength and endurance, balance, and coordination. Lower body muscular strength and endurance were also found poorer than in other countries [25,43]. 

Aerobic exercise capacity is also one of the fundamental components of health-related physical fitness tests. Among many modes of a cardiorespiratory endurance test, the 2-minute step test introduced by Rikli and Jones as part of the Senior Fitness Test in 1999 is easy to conduct in almost any setting [31,35]. In comparison to other studies [31,41,42,45,46], cardiorespiratory endurance was also found to be poor in Nepalese older adults. Sit and reach, one of the popular field-test components of Health Related Physical Fitness Test (HRPFT) to measure flexibility in the lower back and hamstring muscles, was assessed in this study [53]. Female flexibility was similar to that of Korean female older adults and poorer than that of male older adults [40,42].

Some important findings for the participants from Nepal are as follows: a) very poor BMI, even lower than the baseline of the average recommended by the WHO for older adults; b) with increasing age, both genders showed a declining score in muscular strength and endurance, flexibility, and cardiorespiratory endurance with poor results in comparison to other countries; c) in every test except flexibility, males had higher scores with better results in comparison to women from the same age group; d) 72.45% of people were illiterate; e) 78.79% of people did not get regular health check-ups and more of the older adult population lived in rural areas than urban areas. Poor fitness scores compared to other countries shows a lower level of physical activity and exercises in older adults of Nepal. A higher percentage of illiteracy and lack of regular health check-ups suggests a poor awareness status about health and physical activities. Poor BMI values show a higher risk of mortality in older adults of Nepal. Considering the finding of this study: a) Administrators, physical educators, instructors, and health instructors should encourage older adults to do physical activities and exercise as recommended by the WHO [54]; b) Regular awareness programs about physical activities, good health, and quality of life should be providedl c) A wide range of activities should be identified and arranged according to older adults’ interests and needs; and d) Fitness should be tested regularly as per norms. 

Cross-sectional design, covering only health-related fitness components, and testing only in the Terai region of Nepal are the limitations of this study. Despite these limitations, this study included both urban and rural areas of Sub-Metropolitan City and all the major ethnic groups of Nepal. A body composition analyzer (Inbody, Korea) was use to assess BMI and percent body fat. 

The results of this study could be used as approximate indicative values for comparisons among the same health-related fitness scores of older adults from other developing Asian countries similar to Nepal. In addition, the scores can be used as benchmark values for health-related fitness evaluations of older adults of Nepal.

## 5. Conclusions

This study may help the health and physical education sector to develop appropriate physical activities or exercise programs for older adults. It may help to identify the fitness level of an individual and help fitness instructors to develop suitable physical activity interventions for older adults based on their age. In addition, an awards system may encourage individuals to participate in physical activities or exercise continuously, which helps lead to a better and more independent life in upcoming years. Longitudinal studies along with functional fitness studies on older adults and adult fitness studies are necessary in the future.

## Figures and Tables

**Figure 1 ijerph-17-02723-f001:**
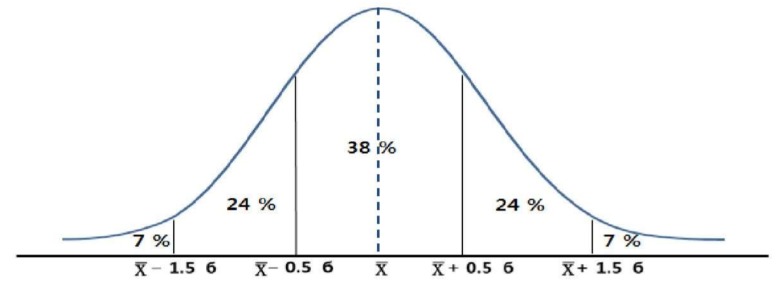
Cajori 5-grade evaluation norms.

**Table 1 ijerph-17-02723-t001:** General characteristics of the participants.

Characteristics	Sub-Characteristics	No. of Participants	Percent (%)
Gender	Male	449	44.50
	Female	560	55.50
Ethnic Group	Brahman	199	19.72
	Chettri	183	18.14
	Janjati	457	45.29
	Dalit	101	10.01
	Madeshi	4	0.40
	Other	65	6.44
Education	Illiterate	731	72.45
	Non-Formal	129	12.78
	Basic	91	9.02
	Secondary	44	4.36
	Under-Graduate	7	0.69
	Graduate	7	0.69
Marital Status	Married	774	76.71
	Unmarried	0	0.00
	Widow/Widower	235	23.29
Family Type	Small with Children	62	6.14
	Small without Children	88	8.72
	Joint	859	85.13
^*^ Tobacco Intake	Non	950	94.15
	Past	9	0.89
	Current	50	4.96
Drinking Habit	No	862	85.43
	Yes	147	14.57
Covered Area	Rural	705	69.87
	Urban	304	30.13
Regular Health Check-up	Within six months	87	8.62
	Within twelve months	127	12.58
	Never	795	78.79

* Tobacco intake include both smoke and smokeless tobacco.

**Table 2 ijerph-17-02723-t002:** Height based on age group and gender, Unit: cm.

Age Group	Male			Female		
	*n*	M	SD	*n*	M	SD
60–64	121	152.9	8.15	198	152.1	8.72
64–69	140	151.3	8.45	185	152.4	8.52
70–74	85	151.4	8.16	101	150.8	8.84
75–79	66	151.2	8.41	50	153.5	7.73
80+	37	148.4	9.12	26	154.0	8.46

M = mean; SD = standard deviation.

**Table 3 ijerph-17-02723-t003:** Weight based on age group and gender, Unit: kg.

Age Group	Male			Female		
	*n*	M	SD	*n*	M	SD
60–64	121	52.5	11.25	188	52.0	11.41
64–69	140	48.5	10.93	185	48.5	11.76
70–74	85	47.6	8.49	101	46.1	10.80
75–79	66	46.8	11.00	50	48.5	10.53
80+	37	43.3	9.87	26	47.0	10.09

M = mean; SD = standard deviation.

**Table 4 ijerph-17-02723-t004:** Body mass index (BMI) based on age group and gender before log transformation, Unit: kg/m^2^.

Age Group	Male			Female		
	*n*	M	SD	*n*	M	SD
60–64	121	21.7	3.59	198	22.9	4.91
64–69	140	20.9	3.65	185	20.9	4.12
70–74	85	20.5	3.05	101	20.6	4.00
75–79	66	19.2	3.92	50	20.2	3.89
80+	37	19.0	3.41	26	19.3	3.00

M = mean; SD = standard deviation.

**Table 5 ijerph-17-02723-t005:** BMI based on age group and gender after log transformation, Unit: kg/m^2^.

Age Group	Male			Female		
	*n*	M	SD	*n*	M	SD
60–64	121	3.06	0.61	198	3.11	0.21
64–69	140	3.02	0.17	185	3.02	0.20
70–74	85	3.01	0.14	101	3.01	0.19
75–79	66	19.2	3.92	50	20.2	3.89
80+	37	19.0	3.41	26	19.3	3.00

M = mean; SD = standard deviation; age groups 60–64, 65–69, and 70–74 are transformed log data.

**Table 6 ijerph-17-02723-t006:** Percent body fat based on age group and gender, Unit: %.

Age Group	Male			Female		
	*n*	M	SD	*n*	M	SD
60–64	121	23.1	7.90	198	34.3	9.95
64–69	140	23.4	7.80	185	32.2	9.07
70–74	85	23.6	7.80	101	31.6	9.46
75–79	66	23.7	7.90	50	31.3	9.24
80+	37	23.8	8.00	26	29.8	8.95

M = mean; SD = standard deviation.

**Table 7 ijerph-17-02723-t007:** Grip strength based on age group and gender, Unit: kg.

Age Group	Male			Female		
	*n*	M	SD	*n*	M	SD
60–64	121	26.9	6.91	198	17.7	4.26
64–69	140	23.0	6.37	185	16.0	4.26
70–74	85	22.4	6.29	101	15.3	4.30
75–79	66	21.4	7.01	50	14.5	4.62
80+	37	17.2	6.13	26	12.6	3.75

M = mean; SD = standard deviation; dominant hand’s score was calculated as mean and standard deviation.

**Table 8 ijerph-17-02723-t008:** Relative grip strength based on age group and gender, Unit: %.

Age Group	Male			Female		
	*n*	M	SD	*n*	M	SD
60–64	121	49.3	12.20	198	36.6	11.03
64–69	140	45.1	12.72	185	36.2	10.65
70–74	85	45.0	12.77	101	35.7	10.44
75–79	66	43.2	13.10	50	34.8	10.59
80+	37	36.3	13.13	26	32.8	10.63

M = mean; SD = standard deviation; relative grip strength was calculated considering the weight.

**Table 9 ijerph-17-02723-t009:** Sit and reach based on age group and gender before log transformation. Unit: cm.

Age Group	Male			Female		
	*n*	M	SD	*n*	M	SD
60–64	121	7.1	5.47	198	14.8	5.81
64–69	140	5.73	5.27	185	14.3	6.43
70–74	85	5.2	5.07	101	13.1	5.5
75–79	66	4.08	3.7	50	10.8	5.9
80+	37	3.98	3.58	26	10	2.10

M = mean; SD = standard deviation.

**Table 10 ijerph-17-02723-t010:** Sit and reach based on age group and gender after log transformation. Unit: cm.

Age Group	Male			Female		
	*n*	M	SD	*n*	M	SD
60–64	121	1.5	1.16	198	2.5	0.80
64–69	140	0.5	0.55	185	2.4	0.83
70–74	85	0.9	1.36	101	2.4	0.73
75–79	66	0.8	1.17	50	2.1	0.95
80+	37	0.8	1.2	26	2.1	0.79

M = mean, SD = standard deviation.

**Table 11 ijerph-17-02723-t011:** One-minute sit-to-stand based on age group and gender, Unit: reps.

Age Group	Male			Female		
	*n*	M	SD	*n*	M	SD
60–64	121	27.3	7.24	198	22.7	7.86
64–69	140	25.5	8.10	185	20.7	7.06
70–74	85	23.4	8.52	101	20.0	5.61
75–79	66	22.8	8.04	50	17.9	6.90
80+	37	21.0	7.09	26	16.0	7.26

M = mean; SD = standard deviation.

**Table 12 ijerph-17-02723-t012:** Two-minute steps based on age group and gender, Unit: reps.

Age Group	Male			Female		
	*n*	M	SD	*n*	M	SD
60–64	121	57.7	18.96	198	44.8	16.39
64–69	140	56.2	19.18	185	42.3	15.74
70–74	85	49.5	18.48	101	41.9	11.85
75–79	66	44.5	18.60	50	37.7	13.90
80+	37	42.5	19.33	26	34.5	13.73

M = mean; SD = standard deviation.

**Table 13 ijerph-17-02723-t013:** BMI 5-grade relative evaluation norms, Unit: kg/m^2^.

Gender	Age Group	Obesity	Over Weight	Normal	Thin	Very Thin
Male	60–64	27.1	23.5~27.1	19.9~23.5	16.3~19.9	16.3
	65–69	26.4	22.7~26.4	19.1~22.7	15.4~19.1	15.4
	70–74	25.1	22.1~25.1	19.0~22.1	16.0~19.0	16.0
	75–79	25.1	22.1~25.1	18.3~22.1	14.3~18.3	14.3
	80+	24.1	20.7~24.1	17.3~20.7	13.9~17.3	13.9
Female	60–64	30.2	25.3~30.2	20.4~25.3	15.5~20.4	15.5
	65–69	27.0	22.9~27.0	18.8~22.9	14.7~18.8	14.7
	70–74	26.6	22.6~26.6	18.6~22.6	14.6~18.6	14.6
	75–79	26.0	21.1~26.0	18.2~22.1	14.3~18.2	14.3
	80+	23.8	20.8~23.8	17.8~20.8	14.8~17.8	14.8

**Table 14 ijerph-17-02723-t014:** Percent body fat 5-grade relative evaluation norms, Unit: %.

Gender	Age Group	Obesity	Over Weight	Normal	Thin	Very Thin
Male	60–64	35.0	27.1~35.0	19.2~27.1	11.3~19.2	11.3
	65–69	35.1	27.3~35.1	19.5~27.3	11.7~19.5	11.7
	70–74	35.3	27.5~35.3	19.7~27.5	11.9~19.7	11.9
	75–79	35.6	27.7~35.6	19.8~27.7	11.9~19.8	11.9
	80+	35.8	27.8~35.8	19.8~27.8	11.8~19.8	11.8
Female	60–64	49.3	39.3~49.3	29.4~39.3	19.4~29.4	19.4
	65–69	45.8	36.7~45.8	27.7~36.7	18.6~27.7	18.6
	70–74	45.8	36.3~45.8	26.8~36.3	17.4~26.8	17.4
	75–79	45.2	36.0~45.2	26.7~36.0	17.5~26.7	17.5
	80+	43.2	34.2~43.2	25.3~34.2	16.3~25.3	16.3

**Table 15 ijerph-17-02723-t015:** Grip strength 5-grade relative evaluation norms, Unit: kg.

Gender	Age Group	Excellent	Very Good	Good	Poor	Very Poor
Male	60–64	37.3	30.4~37.3	23.4~30.4	16.5~23.4	16.5
	65–69	32.6	26.2~32.6	19.9~26.2	13.5~19.9	13.5
	70–74	31.8	25.5~31.8	19.3~25.5	13.0~19.3	13.0
	75–79	31.8	24.8~31.8	17.9~24.8	10.9~17.9	10.9
	80+	26.4	20.3~26.4	14.2~20.3	8.0~14.2	8.0
Female	60–64	24.1	19.8~24.1	15.5~19.8	11.3~15.5	11.3
	65–69	22.4	18.1~22.4	13.9~18.1	9.6~13.9	9.6
	70–74	21.8	17.5~21.8	13.2~17.5	8.9~13.2	8.9
	75–79	21.5	16.8~21.5	12.2~16.8	7.6~12.2	7.6
	80+	18.2	14.5~18.2	10.7~14.5	7.0~10.7	7.0

**Table 16 ijerph-17-02723-t016:** Relative grip strength 5-grade relative evaluation norms, Unit: %.

Gender	Age Group	Excellent	Very Good	Good	Poor	Very Poor
Male	60–64	67.6	55.4~67.6	43.2~55.4	31.0~43.2	31.0
	65–69	64.2	51.5~64.2	38.7~51.5	26.0~38.7	26.0
	70–74	64.2	51.4~64.2	38.6~51.4	25.9~38.6	25.9
	75–79	62.9	49.8~62.9	36.7~49.8	23.6~36.7	23.6
	80+	56.0	42.8~56.0	29.7~42.8	16.6~29.7	16.6
Female	60–64	53.2	42.1~53.2	31.1~42.1	20.1~31.1	20.1
	65–69	52.2	41.5~52.2	30.9~41.5	20.2~30.9	20.2
	70–74	51.4	40.9~51.4	30.5~40.9	20.0~30.5	20.0
	75–79	50.7	40.1~50.7	29.6~40.1	19.0~29.6	19.0
	80+	48.8	38.1~48.8	27.5~38.1	16.9~27.5	16.96

**Table 17 ijerph-17-02723-t017:** Sit and reach 5-grade relative evaluation norms, Unit: cm.

Gender	Age Group	Excellent	Very Good	Good	Poor	Very Poor
Male	60–64	15.3	9.8~15.3	4.4~9.8	−1.1~4.4	−1.1
	65–69	13.6	8.4~13.6	3.1~8.4	−2.2~3.1	−2.2
	70–74	12.8	7.7~12.8	2.7~7.7	−2.4~2.7	−2.4
	75–79	9.6	5.9~9.6	2.2~5.9	−1.5~2.2	−1.5
	80+	9.4	5.8~9.4	2.2~5.8	−1.4~2.2	−1.4
Female	60–64	23.5	17.7~23.5	11.~17.7	6.1~11.9	6.1
	65–69	23.9	17.5~23.9	11.1~17.5	4.7~11.1	4.7
	70–74	21.4	15.9~21.4	10.4~15.9	4.9~10.4	4.9
	75–79	19.7	13.8~19.7	7.9~13.8	2.0~7.9	2.0
	80+	13.2	11.1~13.2	9.0~11.1	6.9~9.0	6.9

**Table 18 ijerph-17-02723-t018:** One-minute sit-to-stand 5-grade relative evaluation norms, Unit: reps.

Gender	Age Group	Excellent	Very Good	Good	Poor	Very Poor
Male	60–64	38	31~38	24~31	17~24	17
	65–69	38	30~38	22~30	13~22	13
	70–74	36	28~36	19~28	11~19	11
	75–79	35	27~35	19~27	11~19	11
	80+	32	25~32	18~25	10~18	10
Female	60–64	35	27~35	19~27	11~19	11
	65–69	31	24~31	17~24	10~17	10
	70–74	28	23~28	17~23	12~17	12
	75–79	28	21~28	15~21	8~15	8
	80+	27	20~27	12~20	5~12	5

**Table 19 ijerph-17-02723-t019:** Two-minute steps 5-grade relative evaluation norms, Unit: reps.

Gender	Age Group	Excellent	Very Good	Good	Poor	Very Poor
Male	60–64	86	67~86	48~67	29~48	29
	65–69	85	66~85	40~59	27~47	27
	70–74	77	59~77	40~59	22~40	22
	75–79	72	54~72	35~54	17~35	17
	80+	72	52~72	33~52	14~33	14
Female	60–64	69	53~69	37~53	20~37	20
	65–69	66	50~66	35~50	19~35	19
	70–74	60	48~60	36~48	24~36	24
	75–79	59	45~59	31~45	17~31	17
	80+	55	41~55	28~41	14~28	14

**Table 20 ijerph-17-02723-t020:** Older adult health-related fitness award norms.

Fitness Award	Award	Award Norms
Older Adult Health-Related Physical Fitness Award Norms	Gold Awards	- Body Composition (BMI and % Body Fat) excluded - All health-related physical components should be at least 70th percentile (above 30%)
	Silver Award	- Body Composition (BMI and % Body Fat) excluded - All health-related physical components should be equal or above 50th percentile
	Bronze Award	- Body Composition should be in the recommended range - All health-related physical components should be equal or above 30th percentile (below 30%)
Body Composition	Option	7% < male % body fat < 35% 16% < female % body fat < 32%
		18 < male BMI < 25 18 < male BMI < 25

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
