# Peer review of "The Study of Health-Related Fitness Normative Scores for Nepalese Older Adults"

_ijerph, 2020, doi:10.3390/ijerph17082723_

Round 1

Reviewer 1 Report

The article is very interesting and is worth publishing in the journal IJERPH. However, I have the following comments:

Abstract:
- no results in the abstract
- what is the percent body fat test?
- no conclusions,
- no characteristics of the study group (e.g. age, number of participants, etc.)

Introduction:
- insufficient literature review, please complete the literature and expand this part of the work.

Material and Methods
- lack of information on tests verifying the normal distribution on which the method of determining norms is based.

Results
- Tables 1-10 have a very poor description that adds nothing to the article. I suggest expanding the description.
- The use of the Cajori method is justified only if the variables are of a normal distribution nature, otherwise the norms will be incorrect. Has the distribution been checked for compliance with the normal distribution? I did not find such information in the text.

Author Response

Response to Reviewer 1 Comments

The article is very interesting and is worth publishing in the journal IJERPH. However, I have the following comments:

Abstract:

Point 1:

no results in the abstract

Response 1:

Thank you for your comments.

It has been added.

Point 2:

what is the percent body fat test?

Response 2:

Thank you for your comments.

Percent body fat or body fat percent is the total mass of fat divided by total body mass and multiplied by 100, which include essential body fat and storage body fat.

Point 3:

no conclusions,

Response 3:

Thank you for your comments.

It has been updated.

Point 4:

no characteristics of the study group (e.g. age, number of participants, etc.).

Response 4:

Thank you for your comments.

 It has been added.

Introduction:

Point 5:

- insufficient literature review, please complete the literature and expand this part of the work..

Response 5:

Thank you for your comments.

Few references have been added and updated.

Material and Methods

Point 6:

- lack of information on tests verifying the normal distribution on which the method of determining norms is based

Response 6:

Thank you for your comments.

It has been added.

Results

Point 7:

- Tables 1-10 have a very poor description that adds nothing to the article. I suggest expanding the description.

Response 7:

Thank you for your comments.

Description has been added.

Point 8:

- The use of the Cajori method is justified only if the variables are of a normal distribution nature, otherwise the norms will be incorrect. Has the distribution been checked for compliance with the normal distribution? I did not find such information in the text.

Response 8:

Thank you for your comments

It has been updated. Furthermore, variables are in normal distribution nature except BMI in 60-64, 65-69 and 70-74 age groups in both genders and sit and reach in all age groups and in both genders. Log transformation has been done for the normality test. Mean and standard deviation of log data are updated in tables 5 and 10.

Reviewer 2 Report

This is an important study providing data on the physical fitness of Nepalese older adults.

Introduction: okay

Methods:

Could the authors please provide the protocol for how the data were collected? How was the sample recruited? The descrition of the sampling startegy is unclear and needs to be re-worked. 

Results:

okay but: table 19-21 could be deleted or made available as additional material, they do not provide any additional new information.

Discussion/Conclusion:

Limitations and strenghts are missing of the study and urgently must be worked out. The authors should try and put the results in context. E.g. How does Nepalese compare to other studies on PF in other countries? Discuss some other statistical methods which could be used e.g. LMS Curve Modeling (Cole) for the percentiles?

Author Response

Response to Reviewer 2 Comments

This is an important study providing data on the physical fitness of Nepalese older adults.

Point 1:

Introduction: okay

Response 1:

Thank you for your comments.

Methods:

Point 2:

Could the authors please provide the protocol for how the data were collected? How was the sample recruited? The descrition of the sampling startegy is unclear and needs to be re-worked.

Response 2:

Thank you for your comments.

It has been added in section 2.

Thirty volunteers were recruited in the combined effort of the Sub-metropolitan city and Sudur Paschimanchal Campus. This study was conducted from the last week of July to the third week of august in 2019AD. Four days of training were provided to volunteers. Ten volunteers were well trained to full fill the questionnaire, three volunteers were well trained for each test component.  A permanent Health assistant for the study period was provided by the Urban Health Centre, Dhangadhi Sub-Metropolitan City. Local health assistants and volunteers were provided by the health post of each ward of the sub-metropolitan city. Local volunteers help to communicate with local peoples with the different mother tongue.

Results:

Point 3:

okay but: table 19-21 could be deleted or made available as additional material, they do not provide any additional new information.

Response 3:

Thank you for your comments.

It has been deleted.

Discussion/Conclusion:

Point 4:

Limitations and strenghts are missing of the study and urgently must be worked out. The authors should try and put the results in context. E.g. How does Nepalese compare to other studies on PF in other countries? Discuss some other statistical methods which could be used e.g. LMS Curve Modeling (Cole) for the percentiles?

Response 4:

Thank you for your comments.

It has been updated.

Round 2

Reviewer 1 Report

The paper has been improved since the authors  take into account all the comments of the reviewers. I recommend the article for publication.